# Digital Twin of a Water Supply System Using the Asset Administration Shell

**DOI:** 10.3390/s24051360

**Published:** 2024-02-20

**Authors:** Salvatore Cavalieri, Salvatore Gambadoro

**Affiliations:** Department of Electrical Electronic and Computer Engineering, University of Catania, Viale A. Doria 6, 95125 Catania, Italy; salvatore.gambadoro@phd.unict.it

**Keywords:** digital twin, industry 4.0, water supply system, asset administration shell

## Abstract

The concept of digital twins is one of the fundamental pillars of Industry 4.0. Digital twin allows the realization of a virtual model of a real system, enhancing the relevant performance (e.g., in terms of production rate, risk prevention, energy saving, and maintenance operation). Current literature presents many contributions pointing out the advantages that may be achieved by the definition of a digital twin of a water supply system. The Reference Architecture Model for Industry 4.0 introduces the concept of the Asset Administration Shell for the digital representation of components within the Industry 4.0 ecosystem. Several proposals are currently available in the literature considering the Asset Administration Shell for the realization of a digital twin of real systems. To the best of the authors’ knowledge, at the moment, the adoption of Asset Administration Shell for the digital representation of a water supply system is not present in the current literature. For this reason, the aim of this paper is to present a methodological approach for developing a digital twin of a water supply system using the Asset Administration Shell metamodel. The paper will describe the approach proposed by the author and the relevant model based on Asset Administration Shell, pointing out that its implementation is freely available on the GitHub platform.

## 1. Introduction

The main goal of the Fourth Industrial Revolution (Industry 4.0) is the definition of increasingly flexible, interoperable, and innovative systems, focusing on a continuous evolution of technologies capable of shifting the management of an asset from the physical domain to the virtual domain [1,2].

A fundamental architectural element within the scope of Industry 4.0 is the digital twin (DT), which is a digital representation of a real asset. Through the digital twin, it is possible to develop a virtual model that allows replicating a facility, with the aim of monitoring, managing, and simulating its operation, thereby enhancing control in testing, analysis, prediction, and risk prevention for sensitive processes [3,4]. A digital twin includes the digital replica of the information representative of the real system to be modeled; an information model is used inside a DT for this purpose. Furthermore, applications using the digital information collected from the real system may be defined inside a DT. Due to the lack of a clear architecture defining the elements contained in a DT, i.e., their roles and their interactions, in the last few years, some standardization activities have been conducted to define an architecture for the digital twin. The ISO 23247 digital twin manufacturing framework is among the main proposals [5]. According to the ISO standard, a digital twin features a layered structure, among which a core entity layer deals with the information model to support the operations and management of a complex industrial system. The top layer is the user entity, which is populated by the specific applications that utilize the digital representation of the system [5].

A water supply system (WSS) is a highly complex facility composed of a set of subsystems that realize the different stages of water management from the point of extraction to the final distribution to users. Introducing a digital twin in the modeling and representation of a WSS represents an innovative methodology for water management. Nowadays, having a system that ensures a safe and efficient water supply, with the goal of maximizing the use of water resources, is essential for achieving sustainable water utilization [6]; the use of a DT in a WSS may allow one to reach this goal. In general, there are many advantages offered by the introduction of a DT inside the management of a WSS [7]; considering, for example, the operations and maintenance procedures typical of a WSS, the use of a DT may allow the following:Supporting the operators to choose the best decisions in real-time by simulating the effect on any operation before taking the action in the real system.Implementing energy-saving strategies (e.g., using a reduced number of pumps when demand falls below a dynamically chosen threshold).Detecting anomalies to establish an advanced maintenance system capable of reducing maintenance costs and downtime to minimize disruptions to end-users.Optimizing the operation of the system and maximizing the quality of the service and the water quality.Developing emergency response plans and simulating the behavior of the system under emergency conditions, e.g., realizing an early warning system against possible contamination into the network.

The Asset Administration Shell (AAS) was introduced in 2016 as a core element of the Reference Architectural Model for Industry 4.0 (RAMI 4.0) [7]. The Industry 4.0 component has been described within RAMI 4.0 as the combination of the asset and its digital representation [8,9,10]. The AAS is presented as the official standard for the digital representation of components within the Industry 4.0 system. The current literature offers a large number of papers about the Asset Administration Shell, which point out that it is a primary component of Industry 4.0 for the digital representation of industrial assets (e.g., [11,12,13,14,15,16,17,18,19,20,21,22,23,24]).

The literature presents many examples that demonstrate the feasibility of the realization of DTs by using the AAS [25,26,27,28]. In particular, in [27,28], the current state of the art about realizations of DTs by the AAS metamodel is presented, proving the relevant feasibility of representing heterogeneous industrial assets. Considering the ISO 23247 digital twin framework, ref. [29] presents the implementation of the core entity through the AAS metamodel.

The purpose of this paper is to provide a methodology for creating a digital twin of a WSS using the Asset Administration Shell. This involves modeling all subsystems of the WSS with the aim of achieving a digitized facility capable of harnessing the opportunities and potential derived from the use of digital twins.

The contribution of this paper will only focus on the representations of properties and operations of a WSS and will not consider the aspect of the applications using the DT. From this point of view, the approach here presented is similar to that proposed in [29], where the AAS has been used to implement only the core entity (i.e., the information model) of the ISO 23247 digital twin framework. Although the realization of applications using the WSS model was considered out of scope, the structure of the AAS information model representing the WSS has been defined and implemented taking into account possible applications that could use the model. The research work here presented is carried on inside a research project funded by the Italian Ministry of Enterprises and Made in Italy. Partners of the project suggested applications of interest; they are mainly related to maintenance operations, supervising and control operations, and water quality assurance. The AAS information model structure defined by the authors includes suitable support for these kinds of applications. For example, in order to ensure an advanced and efficient maintenance system in the water supply system, a specific maintenance model was developed, providing a structured representation of maintenance activities, offering a complete view of the scheduled maintenance plans and historical records of previous maintenance activities. It is important to highlight that different kinds of applications (e.g., energy monitoring and saving) may be easily supported by extending the AAS information model presented in the next sections. The AAS metamodel is highly flexible and features several methods to perform extensions of the information set.

This paper will be structured as follows. Section 2 points out the related works present in the current literature. Section 3 and Section 4 give an overview of the WSS and the AAS, respectively. Section 5 presents the methodology proposed by the authors to realize a DT of the WSS by the AAS. Section 6 highlights the real implementation of the AAS model, which is free and available in a repository on GitHub. Section 7 will present a case study, applying the AAS model here presented to a real WSS. Final remarks will conclude the paper.

## 2. Related Works

In recent years, research in water networks has considered the possibility of implementing DTs to support the relevant management. The introduction pointed out some of the main advantages that may be achieved. In [30], graph convolutional neural network theory linked to hydraulic models is proposed for generating a DT of the water distribution system. In [31], the research presented defines a new methodology for an efficient application of DT expertise within water distribution networks. In [32], a smart water grid with a DT is proposed for a water infrastructure to improve monitoring, management, and system efficiency; such a tool allows live monitoring of system components, which can analyze different scenarios and variables, such as pressures, operating devices, regulation of different valves, and head-loss factors. In [33], a wide set of digital water services suited to support various tasks along with common workflows in the management of water distribution networks are presented; such services are implemented as plugins using the DT of the water distribution network. In [34], the authors propose a new framework based on the DT paradigm aimed to introduce intelligence and autonomy throughout the loop of data acquisition and processing as well as asset control and service generation and delivery.

The introduction pointed out that the current literature presents a huge amount of examples of the successful use of an Asset Administration Shell for the realization of the digital twin of real systems in the Industry 4.0 context. Among the most relevant examples, in [35], the AAS has been used to realize a DT for predictive maintenance purposes. In [27], the feasibility of the AAS has been proved in order to represent heterogeneous industrial assets and their DT; a robotic arm has been considered in the validation of the proposal. The goal of the paper [36] is to validate the use of AAS as a standardized digital representation of an asset during the entire product lifecycle management process of an asset. The authors of the paper [37] use the AAS as a synonym for DT to demonstrate the potential of DT for multi-agent systems using an AAS metamodel.

To the best of the authors’ knowledge, at the moment, the adoption of an Asset Administration Shell for the digital representation of WSS is not present in the current literature. For this reason, the authors believe that the contribution of this paper may be important to introduce a modeling approach based on the AAS for the realization of a digital twin of a WSS. In order to share the proposal here presented with the researchers in this field, the implementation of the AAS model of the WSS has been uploaded on GitHub and is freely available to be downloaded and used; Section 6 will point out the relevant link.

## 3. Water Supply System

A water supply system is a complex system designed to collect, treat, and distribute water in a specific geographical area. Due to the several operations involved, a WSS is composed of different subsystems, each of which has a well-defined role in the water supply chain that enables the provision of both potable and non-potable water to the target area [38]. As shown by Figure 1, a WSS features one or more raw water sources, each of which is a source of untreated water (such as a river or lake) and serves as the starting point of the supply system. Water is extracted from the raw water source through a raw water pumping facility. This facility includes intake structures (not shown in Figure 1) in charge of extracting water from the source to convey it to the water treatment system.

A water treatment facility allows making the raw water safe for human use. The treatment procedure may be made up of several steps, starting with coagulation and flocculation, involving the addition of chemicals such as coagulants and flocculants to the raw water. The second step is sedimentation, where water is collected in large tanks designed to separate sediments from the water. The third phase is filtration, where water passes through filters to remove smaller particles, bacteria, viruses, and other impurities. The last two stages are disinfection and softening, which are responsible for eliminating viruses and bacteria and reducing water hardness, respectively [39]. Some water supplies may contain radionuclides (small radioactive particles), specific chemicals (such as nitrates), or toxins (such as those made by cyanobacteria); specialized methods to control or remove these contaminants can also be part of water treatment [40].

Untreated water achieved by the raw water pumping facility may be collected in storage tanks before undergoing treatment. Sometimes, some of the treatments described before may be realized locally in the tanks.

Once the treatment is complete, the water reaches the water distribution system, which is responsible for the final stage of the water supply, where the treated water is provided to end consumers. The water distribution system consists of subsystems that handle the pumping and storage of water, which is then distributed through piping systems [38].

As shown by Figure 1, water is transported from one facility to another through water transmission systems made up of complex networks of pipes, pumps, and valves that enable control over the water flow.

## 4. Asset Administration Shell Metamodel

In the context of Industry 4.0, every real system, i.e., asset, is overseen through an Asset Administration Shell (AAS) metamodel that contains all relevant information about it. This encompassing data includes its properties and operations, as well as documentation, datasheets, CAD files, or source code. To effectively structure the information within the AAS, a consistent and unique method is required. The document titled “Details of the Asset Administration Shell” [41] offers an AAS metamodel to fulfill this requirement. At a higher level, an AAS metamodel is composed of a header and a body, as illustrated by Figure 2.

The header contains essential information for identifying both the asset and the AAS, while the body encompasses all the asset’s properties and operations. These are organized within submodels, each focusing on specific aspects of the asset, such as energy efficiency or positioning.

### 4.1. AAS Metamodel Common Classes

Entities in the AAS metamodel can inherit from more than one common class. Common classes are abstract classes used to describe aspects shared by metamodel entities. From a practical point of view, common classes collect attributes that can be shared by different classes in the metamodel.

Some of the main common classes defined in the AAS metamodel and used in the proposal here presented are Referable, Identifiable, and HasSemantics.

All entities in the metamodel inheriting from the Referable class provide a short identifier (idShort) that is unique only in the context of its namespace.

Identifiable entities, instead, consist of all those classes whose instances can be uniquely and globally identified by means of their attribute identification. Identification may belong to one of the following types: IRI (URI), IRDI, and Custom. These identifiers will be used inside instances of the class ReferenceElement (or Ref, for brevity), as will be discussed in the next subsection.

The entities inheriting from HasSemantics identify all those classes that can be described by means of a concept. One of the core entities of the AAS metamodel to achieve interoperability is ConceptDescription; it is used to define the semantics of entities inside the AAS metamodel. Every element in AAS that is HasSemantics should have its semantics described by a ConceptDescription. ConceptDescription should follow a standardized template to describe a concept. The only templates available in the metamodel are used to define semantics according to IEC 61360 [42]. The class HasSemantics defines an attribute SemanticId that is a reference pointing to a ConceptDescription.

### 4.2. ReferenceElement

The AAS metamodel introduces an important referencing mechanism to establish relationships among the entities that make up the AAS. This mechanism relies on the ReferenceElement (Ref) class, which plays a fundamental role in creating connections and relationships between the elements constituting the digital representation of a system. Through the ReferenceElement class, structured navigation within the AAS becomes possible. For example, if one submodel refers to another through a ReferenceElement, it is possible to follow this link to access the related information. The relationships established through ReferenceElements can dynamically change over time; this dynamism is highly requested in the context of Industry 4.0, where information may undergo changes during operations or due to events in the industrial environment. The relationship is realized through an attribute of this class, named value, which contains the identification of the entity pointing that must be Identifiable. For more information on the ReferenceElement class, refer to [41].

### 4.3. Class Hierarchy in the AAS

All the classes outlined in the AAS metamodel serve the purpose of breaking down and simplifying the representation of the internal information within an AAS, as detailed in [43]. This hierarchical structure aligns with the intended organization of the AAS, with the highest-level class, AssetAdministrationShell, representing the AAS in its entirety. The Submodel class, on the other hand, embodies a specific facet of the asset, while the abstract class SubmodelElement encompasses all elements that should be grouped within a ubmodel, such as properties, operations, files, and more.

Figure 3 shows (using the UML formalism) the class hierarchy in AAS metamodel, limiting to Submodel class; the same figure presents a legend to recall the meaning of the UML formalisms used. As illustrated, a Submodel class establishes a composition relationship with the SubmodelElement class. Composition means that the Submodel class contains multiple instances of the SubmodelElement class, making the ubmodel a composite entity made up of multiple SubmodelElement entities.

A SubmodelElement serves as an abstract superclass encompassing all entities responsible for shaping the internal structure of a submodel, such as properties, files, and operations.

A data element is a SubmodelElement that is not further composed out of other SubmodelElements; the data element has a value or a predefined number of values like range data. The Property class defines attributes for holding data values and specifies the data value’s type. Properties stand as vital elements within a submodel since they constitute the primary source of information concerning an asset.

Lastly, the class ReferenceElement, introduced in the previous subsection, defines a logical reference to another element within the same AAS or a different one; it can also represent a reference to an external object or entity.

In a broader context, a SubmodelElement has the capacity to encompass other SubmodelElements, forming an internal hierarchy within itself. The concrete class SubmodelElementCollection (SEC) serves a crucial role in this context, as it is defined as a set or list of SubmodelElements. Such a collection can be organized in order and may permit or prohibit duplicate elements. SEC holds significant importance as it is the sole entity that enables the internal structuring of a submodel, which is akin to a folder within a directory.

### 4.4. Submodel Template

In the Asset Administration Shell, Submodel Templates allow for establishing a standardized and consistent structure for submodels within an AAS. These templates help to ensure coherence and compliance when creating submodels for similar or related assets. As described in [44], a Submodel Template defines the internal structure of a submodel, including the classes, properties, attributes, and relationships that should be present. They can be reused for similar or identical assets. Despite the standard definition provided by the Submodel Template, it is possible to customize the submodels according to specific needs. This allows for adapting the standard model to a particular asset without having to create it from scratch. The use of Submodel Templates can enhance interoperability between different systems and organizations since they follow a common structure, enabling data to be exchanged more effectively and comprehensibly.

For the implementation of the digital twin of the WSS here presented, several Submodel Templates were used. They were developed by the Industrial Digital Twin Association (IDTA), which provides developers with a list of standardized and ready-to-use Submodel Templates [45]. The Submodel Templates from IDTA used in the implementation of the AAS in this project are as follows:Submodel Template Identification: This has been designed to represent the identification information of an industrial asset; it provides a standardized framework for uniquely identifying an asset within the AAS environment and describing its identification properties. This submodel can include several properties to comprehensively represent the identification information of the asset, such as a unique ID, the type of asset, the asset’s name, a detailed description, categories or classifications (such as industrial sector, application, and functional area), physical location within a facility or structure, and contact information for individuals or entities responsible for the asset (such as owner’s name, affiliated organization, phone number, and email address). These identification details are very important for distinguishing and unequivocally referring to an asset within the AAS ecosystem, facilitating accurate identification, traceability, and management of industrial resources.Submodel Template Documentation: This has been designed to represent documentation associated with the manufacturer for an industrial asset. This submodel facilitates management and provides access to a range of documents supplied by the asset’s manufacturer, such as instruction manuals, installation guides, certificates of compliance, drawings, technical diagrams, and other relevant information. Within the submodel, several properties and relationships are considered to comprehensively represent the manufacturer’s documentation. These include a list of specific documents associated with the asset, featuring details such as name, description, format, creation date, and other information [44]. Version 1.0 has been used.Submodel Template Technical Data: This has been designed to represent the technical data of an industrial asset. This submodel allows for a detailed description of the technical features and specifications of the asset, providing important information for its proper management, maintenance, and usage. The submodel can include several properties and relationships to comprehensively represent the technical data of the asset, such as specific characteristics (e.g., dimensions, weight, capacity, speed, power, accuracy, and expected lifespan), detailed technical specifications (e.g., power supply voltage, operating temperature range, tolerances, and expected performance), and attachments of diagrams, schematics, and technical drawings [44]. Version 1.1 has been used.Submodel Template Boom_Aggregate: This contains information regarding the structure of an object or product as part of a “Bill of Materials” (BOM), which is a list of all components and their relationships within a system or product. It describes the hierarchical structure of the physical objects that make up a specific asset or product. The submodel defines key elements, including the unique identification of components, the hierarchical structure highlighting parent–child relationships, a detailed description of each component, the quantity required to assemble the asset, and links to other submodels within the AAS to establish connections with related information [46]. Version 1.0 has been used.

## 5. Asset Administration Shell Model for the Water Supply System

The aim of this section is to provide a comprehensive description of the AAS model developed by the authors to represent a water supply system. As outlined in Section 3, a WSS is a complex system comprising at least five subsystems: raw water pumping facility, water transmission system, raw water storage, water treatment facilities, and water distribution system. In most cases, these subsystems may be treated as self-contained entities, and each of them is represented by its own Asset Administration Shell model. More complex subsystems may share the same basic components (e.g., pumps, pipes); in this case, the relevant AAS model will point (through ReferenceElements) to the AAS model representing each basic component.

In the following subsections, a detailed description of the AAS models representing the basic components will be given; then, the description will focus on the complex subsystems. Finally, the overview will focus on the AAS modeling of the entire water supply system.

### 5.1. AAS Models of Basic Components

As mentioned, the aim of this subsection is to provide an overview of the main features of the AAS models that were defined to represent the basic components that make up the complex systems of the WSS.

Each AAS model is made up of several submodels, some of which are common to them. In particular, Templates Identification, Documentation, and TechnicalData Submodels (described in Section 4.4) are present in every AAS model representing a basic component. The other common submodels are described in the following.

The Sensors Submodel contains detailed information related to the sensors connected to each basic component. Each sensor is associated with a SubmodelElementCollection that not only provides a comprehensive technical description, highlighting the specific characteristics of the sensor, but also includes additional data such as the manufacturer and serial number, providing a complete overview of its identification specifications. Within each SMC associated with the sensor, there is another SMC named “Data” that contains telemetric data produced by the sensor. Among the information associated with each data point, there are timestamp values, relevant units of measurement, and the type of detected data, which are all accompanied by a detailed description.

The Capabilities Submodel defines the operational and functional capabilities of each basic component. This submodel allows the descriptions of the relevant features and abilities, enabling external systems to understand what the asset can do and how it can be used within its applied context.

The Operations Submodel is designed to define and manage the operations that can be performed on the component itself. It contains detailed information about the functions that can be carried out on the component and the conditions or prerequisites associated with these operations.

The Maintenance Submodel is designed to digitally represent information related to the maintenance activities of a component within the AAS. This type of submodel provides a structure for the management and monitoring of maintenance operations, such as the current status or historical data of maintenance, contributing to the optimization of maintenance for industrial assets.

The Pipes Submodel contains ReferenceElements to the AAS model representing pipes to which a basic component is connected. This submodel plays a very important role as it realizes a digital registry that lists all the physical connections between the model representing the basic component and the system’s pipes. The Pipes Submodel is present in every AAS model of the basic components, except in the AAS modeling of the pipe and the AAS modeling of the water pump driver, as it is not connected to any pipe.

#### 5.1.1. AAS_Pipe

This AAS model represents a pipe and contains information about the physical features, operating conditions, maintenance, monitoring, and documentation.

In addition to the common submodels described before, the AAS_Pipe features the Connections Submodel, which allows tracking of the elements through which water enters and exits the pipe. The Connections Submodel is structured through the use of SubmodelElementCollection. Each SMC represents a connection containing at least two ReferenceElements pointing to the source asset, from which the water comes, and the destination asset, to which water is forwarded.

Figure 4 shows an example that helps to better understand the meaning and the use of the Connections Submodel. As it can be seen, the real system to be modeled is made up of a pipe connected to two tanks. The same figure shows the AAS models used to represent these components; the AAS_Tank model will be described in the next subsection. Considering the AAS_Pipe1 model, it represents the instance of the AAS_Pipe model describing the real pipe Pipe1. The Connections Submodel for this instance presents the SMC Connection1, containing two ReferenceElements pointing to the model of the source tank (i.e., Tank1) and to the model of the destination tank (i.e., Tank2).

#### 5.1.2. AAS_Tank

This AAS model includes information about the basic component tank, e.g., its dimensions, location, operating conditions, and maintenance. A tank in a WSS can be used not only to contain water but also to hold other substances, such as chemicals, especially during the water treatment phase. For this reason, the TechnicalData Submodel will contain specific information to describe the content and use of the tank.

In AAS_Tank, the Pipes Submodel described at the beginning of Section 5.1 is present. Figure 4 allows a better understanding of the role of this submodel. AAS_Tank1 and AAs_Tank2 are two instances of the AAS_Tank type, modeling the two tanks shown in Figure 4. It is clear how, through the set of ReferenceElements, the model representing each tank can access the AAS modeling the pipe connecting the tanks, and, conversely, from the AAS representing the pipe, it is possible to access to the digital representations of the tanks.

#### 5.1.3. AAS_Valve

This AAS model is a comprehensive digital representation of a physical valve within a WSS. It provides data on the valve’s type, position, operational parameters, maintenance history, and connections, facilitating efficient management and control of the valve’s functions and performance in the system.

#### 5.1.4. AAS_WaterPump

This model (AAS_WP for short) allows for the representation of one of the most found basic components in a WSS, i.e., the water pump. It holds telemetric and operational information, data about the pump’s type, location, operational parameters, maintenance history, and connections that enable efficient management of pump usage.

#### 5.1.5. AAS_PumpDriver

This AAS model (AAS_PD for short) is a detailed digital representation of the electronic or mechanical device used to control a pump within a WSS. It includes information about the driver’s type, control parameters, connections, and other relevant specifications, enabling effective management and monitoring of the driver to ensure the proper operation of the pump.

#### 5.1.6. AAS_ModularComputingSystem

The AAS_ModularComputingSystem (AAS_MSC for short) allows for the representation of a modular computing system such as PLC (Programmable Logic Controller) and RTU (Remote Terminal Unit). A modular computing system refers to a system composed of interconnected modules, among which at least a module with computing capabilities must be present (e.g., a module with CPU and RAM); other modules may compose the system, among which communication modules are needed for the I/O data exchange. Within this context, the AAS model maintains a detailed digital representation of each module of the system.

In addition to the common submodels described before, this AAS model features the Modules Submodel, which is designed to digitally represent modules composing the system. Considering modules responsible for managing input and output data exchanges, the structure of the submodel includes an SMC object called “IOModules”, which is further organized through two other SMCs, namely, “Input” and “Output”. Within Input SMC, there is the representation of modules that handle incoming information, while in Output SMC, modules that produce outgoing information are included. Each module is represented through another SubmodelElementCollection containing information such as its properties (e.g., type of I/O module, technical specifications) and available channels. Figure 5 illustrates the structure of the Module Submodel.

As can be seen in Figure 5, each module contains properties (Pr) and information about the channels present (i.e., the real input or output connection present in the module). Each channel is represented by an SMC containing several properties and a ReferenceElement pointing to the representation of the component to which the channel is connected. If the channel is an input, a ReferenceElement SourceChannel will point to the AAS modeling the real input; if the channel is an output, a ReferenceElement DestinationChannel will point to the AAS modeling the real output.

#### 5.1.7. AAS_CommunicationSystem

The AAS_CommunicationSystem (AAS_CS for short) represents a communication system between machines, sensors, actuators, and other devices within an industrial environment; an example of a communication system may be represented by an ethernet-based switch. The AAS model includes detailed information about communication configurations, data flows between system components, communication interfaces, and information about communication protocols adopted.

In addition to the common submodels already described, there is the LogicalConnection Submodel.

The LogicalConnection Submodel is designed to maintain information about connections with devices attached to the communication system. Each communication channel within the communication system (e.g., I/O ports in a switch) is represented by an SMC that provides detailed information on the relevant features, including security, service quality, connections to the physical device, and the communication protocol used in the communication with the device. These aspects are represented by properties and by ReferenceElements. Figure 6 illustrates the structure of the LogicalConnection Submodel.

As can be seen, for each available communication channel, information is given through properties and references. In particular, ReferenceElement Device points to the AAS modeling of the device attached to the channel, and the ReferenceElement DestinationChannel allows for pointing to the AAS model of the component describing the communication channel of the device, which is involved in the communication; ReferenceElement DeviceProtocol points to the AAS describing the communication protocol used. In the next subsection, an example of this AAS model will be presented in order to better explain the relevant contents.

#### 5.1.8. AAS_CommunicationProtocol

The AAS_CommunicationProtocol (AAS_CP for short) allows for the representation of each communication protocol used in the system. This AAS includes information about the configuration and parameters required to implement the protocol.

The Parameters Submodel has been defined to manage the configuration needed for the implementation of the communication protocol. Configuration is given by a set of parameters that depend on the specific protocol used. For example, assuming the use of a Profinet protocol [47], among the parameters that can be managed there are the following: IP address, SubnetMask, SendClock, and UpdateTime.

In order to better understand the representation of this AAS, the example has been prepared and is shown in Figure 7; it includes the AAS_CommunicationSystem and AAS_ModularComputingSystem models, allowing to better clarify their role and usage.

As can be seen, two PLCs are connected to each other by an ethernet-based switch using the Profinet communication system. Figure 7 shows that the information flow goes from PLC1 to PLC2, i.e., it is an unidirectional information flow.

Figure 8 shows the relevant representation using the AAS_ModularComputingSystem, AAS_CommunicationSystem, and AAS_CommunicationProtocol models.

The two PLCs, referred to as PLC1 and PLC2, are presented as instances of the AAS_ModularComputingSystem. The Modules Submodel describes the available I/O modules; among them, an output module is present in PLC1, and an input module is present in PLC2 (they feature the same name, Module1). Inside both Module1 SMCs, Channel0 is used for the data exchange from PLC1 to PLC2.

The switch shown in Figure 7 is modeled by an instance of AAS_CommunicationSystem; it is named Switch1. The Submodel LogicalConnections allows for a representation of the connectivity with the two PLCs. In particular, Channel0 of the switch is connected to PLC1 (by the ReferenceElement Device pointing to PLC1 AAS). The Channel1 of the switch is connected to PLC2 (by the ReferenceElement Device pointing to PLC2 AAS). The ReferenceElements DestinationChannel present in the LogicalConnection Submodel allows a representation of the information flow; it is clear that information flows from Channel0 of the switch to Channel1 of the same switch and from Channel1 of the switch to the Channel0 of the PLC2. In the PLC1 AAS, the ReferenceElement DestinationChannel points to the Channel0 of the Switch1 AAS, representing the information flow from the PLC1 to the switch.

The ReferenceElements DeviceProtocol present in the Switch1 AAS LogicalConnection Submodel allows for a representation of the protocol used in the communication and the relevant parameters. As can be seen, the references point to the AAS named Profinet1 and Profinet2, which are instances of the AAS_CommunicationProtocol.

The AASs Profinet1 and Profinet2 contain a Parameters Submodel, which is able to represent parameters of the Profinet protocol used for the exchange of information between PLC1 and PLC2, among which are the device’s IP address, subnet mask, and device role, thus promoting a comprehensive understanding of the configuration and technical aspects of the system.

### 5.2. AAS of Complex Systems

In this section, the AAS models relevant to complex systems of the WSS made up of basic components and/or other complex systems will be described.

The AAS of the complex systems has been defined as a composite AAS, containing references to other AASs, each modeling basic components or other complex systems. In order to realize the composite AAS, a particular submodel has been defined in every AAS, named Components. The main feature of this submodel is to organize ReferenceElements pointing to the AAS models of the components present in the WSS; the organization has been realized through the use of SubmodelElementCollections.

The strength of the Components Submodel lies in its flexibility and adaptability based on the usage scenario. Indeed, with variations in the components present in the WSS, we can modify and customize the Components Submodel to point to specific components of interest. This feature allows for considerable versatility, enabling the submodel to be tailored to the specific needs and configurations of the system, ensuring efficient and customized management of components in different operational contexts.

#### 5.2.1. AAS_PumpingSystem

The AAS_PumpingSystem allows for the representation of a pumping system; it is featured by the presence of several pumps that operate synergistically to coordinate the flow of water within the system or part of the system.

The submodels of the AAS_PumpingSystem include the following: Identification, Documentation, Technical Data, BoM_Aggregate, and Components.

Figure 9 points out the Components Submodel; as it can be seen, it has been defined as a structured collection of SMCs, each of which is made up of one or more ReferenceElements, pointing to the various components of the pumping system.

#### 5.2.2. AAS_RawWaterStorage

This AAS (AAS_RWS for short) provides a detailed description of the WSS subsystem responsible for collecting raw water. As mentioned, the primary goal of the raw water storage system is to gather water in its natural state, without significant treatments.

This AAS features the submodels Identification, TechnicalData, Documentation, BoM_Aggregate, Components, which was already described, and WaterQuality, which is described below.

The WaterQuality Submodel is responsible for storing a set of parameters representing water quality. These parameters include contaminant levels, the presence of chemicals, pH, temperature, and other relevant indicators. Data are collected through dedicated equipment distributed along the entire raw water storage.

#### 5.2.3. AAS_RawWaterPumpingFacility

AAS_RawWaterPumpingFacility (AAS_RWPF for short) is a model designed to represent information related to raw water pumping facilities, focusing on the type of water source used (e.g., lakes, rivers, aquifers, etc.), the associated intake structure, and the components present in the system.

The submodels integrated into the AAS_RawWaterPumpingFacility are as follows: Identification, TechnicalData, Documentation, BoM_Aggregate, and Components, as described previously; a novel submodel, IntakeStructure, has been defined for this AAS.

The IntakeStructure Submodel is designed to include a detailed set of information about the physical configuration and placement of the intake structure within the WSS. The information contained in this submodel includes geographical coordinates of the intake structure, the depth of the structure, the construction material used, implemented wildlife protection systems, and the specific source of raw water from which it is extracted.

Figure 10 shows the structure of the AAS_RWPF, pointing out the Components Submodel.

#### 5.2.4. AAS_WaterTransmissionSystem

The AAS_WaterTransmissionSystem allows for the modeling of water transmission systems. Its submodels include Identification, Technical Data, Documentation, BoM_Aggregate, Components, and WaterQuality, which were already described.

As subsystems such as pumping stations and raw water storage containers are frequently located along the water supply system to pump water from point A to point B or to store it during the transportation process, the Components Submodel is designed to account for their presence by considering them as potential components of the transportation system.

#### 5.2.5. AAS_WaterTreatmentFacility

This AAS manages information related to water treatment plants. A water treatment facility consists of various treatment stages, including coagulation, flocculation, sedimentation, and others. For each of these phases, a submodel has been created to contain detailed characteristics and data from sensors installed in the system.

Therefore, in addition to the submodels we have already seen presented for the other AASs (i.e., Identification, TechnicalData, Documentation, BoM_Aggregate, Components, and WaterQuality), the authors have defined the following submodels for each of the water treatment stages: Pretreatment, Coagulation, Flocculation, Sedimentation, Filtration, Disinfection, and Nitrate Filter. These submodels enable comprehensive and timely management of each stage of the treatment process. For example, in the case of coagulation, the relevant submodel holds specific information about the reagents used, process conditions, and data collected by dedicated sensors. The integration of sensors at every stage of the process enables continuous and real-time analysis of water characteristics, ensuring precise monitoring of plant performance.

#### 5.2.6. AAS_WaterDistributionSystem

This AAS allows for modeling one of the most complex subsystems within a WSS, i.e., the distribution system. It comprises an intricate system of interconnected pipes that transport water from storage tanks to areas where it will be used. This network can span vast territories, connecting different communities and supplying water to huge amounts of consumers. Considering the broad scope of a distribution system, its components are grouped into three parts: the pumping system, responsible for lifting and transporting water, the distribution network that manages the distribution phase, and the storage systems.

Within this AAS, the Components Submodel is structured to separately manage the pumping station, distribution piping, and storage. For the pumping station, the AAS_PumpingSystem described in Section 5.2.1 is used. Therefore, there is a SubmodelElementCollection “PumpingSystems” containing one or more ReferenceElements to AAS_PumpingSystem models.

Regarding distribution piping, a SubmodelElementCollection organizes ReferenceElements related to pipes and valves in the system to model the water distribution phase. For distribution storage, the same approach as distribution piping has been adopted, creating a SubmodelElementCollection that includes tanks, pipes, and valves.

Two SubmodelElementCollections, ModularComputingSystems and CommunicationSystems, are also present to manage modular computing systems and communication systems of the system, respectively. Figure 11 illustrates the structure of this AAS, pointing out only the Components Submodel.

### 5.3. AAS_WaterSupplySystem

The AAS_WaterSupplySystem (or AAS_WSS for short) is the highest-level system that enables the digital representation and advanced management of all key elements within the water supply system, providing a unified and interoperable view of all assets involved, thus facilitating their monitoring, management, and maintenance. The AAS_WSS includes detailed information about each asset within the system, pointing to the AAS models representing the subsystems that make up the supply system. Similarly, it points to AAS models representing control and communication devices within the system, which may be shared by multiple subsystems, allowing for centralized and optimized operations management.

Three novel submodels have been defined for this AAS, i.e., the SubSystems, ModularComputingSystems, and CommunicationSystems Submodels.

The SubSystems submodel provides an organized framework for the management and interaction with subsystems constituting the water supply system. It is based on the same idea used for the Components Submodels, i.e., it is structured using multiple SubmodelElementCollections that organize ReferenceElements, allowing them to point to the AAS of each subsystem within the water supply system. The Subsystems Submodel serves as a key entry point to explore and understand the structure, status, and dynamics of each subsystem.

The ModularComputingSystems Submodel is designed to represent modular computing devices present in the WSS with the aim of coordinating the computing devices used in one or more subsystems. Within it, ReferenceElements are implemented as direct links to the AAS_ModularComputingSystem models representing each single computing device that must be coordinated.

Similarly, the CommunicationSystems Submodel is dedicated to managing the communication between sensors, actuators, and devices inside the WSS. Also, in this case, it is used when the several communications systems present in each subsystem are integrated and connected to each other. For this reason, it includes ReferenceElements pointing to the AAS_CommunicationSystem models describing communication systems present in the WSS. Figure 12 gives an overview of the AAS_WSS structure, pointing out only the SubSystems Submodel.

## 6. Implementation of the Asset Administration Shell of the Water Supply System

The AAS of the water supply system described in the previous sections has been implemented using the AASX Package Explorer [48]; the tool is provided by the Industry 4.0 consortium and enables the implementation of a digital twin based on the AAS metamodel.

The AAS_WSS model presented here has been uploaded to GitHub and is freely available [49].

Figure 13 shows a screenshot of the implementation of this AAS. For more information about implementation and in order to download the model, the readers may access the repository specified previously.

The previous section presented the main elements composing the submodels designed for WSS. The definition of semantics played a very important role in the definition of the submodels because a mandatory requirement for interoperability is that the meaning of each element (i.e., what each element represents) must be clearly understood by all the partners of the value chain. When a WSS Submodel is explored, it must be clear what each element represents.

For this reason, each element inside the WSS Submodels was defined as HasSemantics (i.e., an entity that inherits from HasSemantics common class); the relevant SemanticId attribute of each element composing the submodels references a semantic description in a ConceptDescription entity defined by the authors following the IEC61360 standard [42]. The ConceptDescription contains the semantic of all the definitions related to the WSS present in each entity of the AAS metamodel. The AAS_WSS model available on GitHub [49] contains the ConceptDescriptions defined by the authors to define the semantics; the reader may explore the several definitions realized by the authors.

## 7. Case Study

The aim of this section is to present an application of the AAS-based modeling of the water supply system proposed in this paper. A real WSS was considered, and the relevant AAS model was defined.

A water supply system realized in one of the cities belonging to the Piemonte region in northern Italy was considered. This system allows for the distribution of water to a population of approximately 60,000 residents with a network of pipelines spanning about 230 km.

Due to the complexity of this WSS, the case study was limited to a subsection of the real system. This subsection is shown in Figure 14; it is made up of raw water sources, raw water pumping facilities, and raw water storage. It is important to point out that although the treatment system is not shown in the figure, a subset of the treatment system is included in the raw water pumping facilities as described in the following.

As shown by Figure 14, the real WSS is made up of three raw water pumping facilities (RWPF1, RWPF2, and RWPF3), ten pumping systems, and eleven raw water storage units (one of them is shared between RWPF1 and RWPF2, as shown). Figure 14 points out the presence of an adduction system that provides water in addition to the three raw water pumping facilities.

In the following subsections, details of the structure of the WSS will be given, and the relevant AAS model for each subsystem will be introduced.

### 7.1. Raw Water Storage

The AAS_RawWaterStorage type described in Section 5.2.2 was used to model each raw water storage; 11 instances of this type have been created, with each representing a single raw water storage system.

In the scenario considered, each raw water storage system is made up of only one tank. This component has been modeled through AAS_Tank, described in Section 5.1.2, including information such as capacity, height, and tank type, in the TechnicalData Submodel. Through the Identification Submodel, each tank was uniquely identified, with additional information about its actual location. The Documentation Submodel allows all the available documentation related to each tank to be attached. Through the Capabilities and Operations Submodels, operations that can be performed on a tank (e.g., opening the tank valve for water outflow and closing it) were described and implemented. The Sensor Submodel describes information about each sensor connected to the tank and maintains telemetry data from the sensors. Finally, through the Pipes Submodel, a record is kept of all pipes connected to the tank.

The pipes connected to the raw water storage were modeled through the AAS_Pipe described in Section 5.1.1.

Modeling of the valves present in each raw water storage system was conducted through the AAS_Valve described in Section 5.1.3. The main features of each valve were represented; in particular, through the Pipes Submodel, it is possible to maintain information about the actual connections of each valve to the existing pipe.

Each raw water system includes a PLC for the control of processes and a switch for the data exchange with other assets. In order to model these systems, the AAS_ModularComputingSystem described in Section 5.1.6 and the AAS_CommunicationSystem described in Section 5.1.7 were, respectively, used.

Through the Components Submodel, it is possible to describe the components described above, i.e., pipes, valves, tanks, communication systems, and computing systems.

Through the WaterQuality Submodel, it is possible to describe and monitor the water quality relevant to each raw water system.

Figure 15 shows the implementation of the AAS model for RawWaterStorage8, pointing out only some of the existing submodels. Considering the Components Submodel, Figure 15 shows the references to the AAS models of the three pipes connecting the RawWaterStorage8 to the pumping systems shown in Figure 14 (i.e., Pumping Systems 7, 9, and 11); furthermore, the submodel presents the references to the AAS models of the three valves, one for each pipe. References to the AAS modeling of the tank and the communication and computing systems are also shown in Figure 15, together with the features related to the water quality.

### 7.2. Water Transmission System

To model the transport system, instances of the AAS_WaterTransmissionSystem type presented in Section 5.2.4 have been created. Each instance includes references to the pumping systems in the Components Submodel, as the WSS considered in this case study features ten pumping stations located along the water transport system; each pumping station has been modeled using the AAS_PumpingSystem type (see Section 5.2.1). Furthermore, references to raw water storage systems, valves, pipes, and computing and communications systems are present in the Components Submodel, as explained in Section 5.2.4 (see Figure 11).

In the following, only the instance of the AAS_WaterTransmissionSystem relevant to the transport system between PumpingSystem5 and RawWaterStorage9 (see Figure 14) will be shown. Figure 16 shows this instance named “WaterTransmissionSystem5”, pointing out the content of the Components Submodel.

The SMC “PumpingSystems” contains a reference to the AAS modeling the PumpingSystem5. The model has been realized as an instance of the AAS_PumpingSystem, as outlined in Section 5.2.1. Figure 16, in the center, shows the instance of the AAS_PumpingSystem representing PumpingSystem5. It features four pumps with a combined flow rate of 35 L. The pumps were modeled through four instances of the AAS_WaterPump type, which is described in Section 5.1.4. Each pump is uniquely identified through the information contained in the Identification Submodel, including details about the manufacturer and the pump’s serial number. Furthermore, the TechnicalData Submodel retains essential technical information, such as the pump type and electrical parameters. Another important aspect pertains to the information kept in the Maintenance Submodel. The pumps within the PumpingSystem5 undergo periodic maintenance procedures, such as the checking of the lubrication oil. All information regarding these procedures, such as component replacements or oil changes, is documented in the Maintenance Submodel. Figure 16, on the right, shows only one instance of the four pumps, i.e., “Pump1PS5”, pointing out some of the relevant submodels. Considering the “PumpingSystem5” instance (shown in the center of Figure 16), the Components Submodel shows the pointer to the four instances of the AAS_WaterPump type.

For each instance of the AAS_WaterPump type, a corresponding instance of the AAS_PumpDriver type (described in Section 5.1.5) has been created to model the control device in charge of managing the pump’s operation. Figure 16, in the center, shows only the references to these instances inside the Components Submodel of the “PumpingSystem5” instance.

The basic components valves and pipes, which are relevant to each transport system considered here, have been defined as instances of the AAS_Valve type, described in Section 5.1.3, and the AAS_Pipe type, described in Section 5.1.1, respectively. Figure 16 shows the references to these instances (on the left) inside the Components Submodel of the “WaterTransmissionSystem5” instance.

Each water transmission system in this WSS case study features the presence of modular computing systems for component control and communication systems that enable the transfer of information with other assets. For modeling these systems, the AAS_ModularComputingSystem described in Section 5.1.7 and the AAS_CommunicationSystem described in Section 5.1.8 were, respectively, used; the relevant references are shown in the Components Submodel of the “WaterTransmissionSystem5” instance, as shown by Figure 16.

### 7.3. Raw Water Pumping Facility

Among the three raw water pumping facilities shown in Figure 14, the RWPF1 is the most complex system, so the authors will focus on it in the following. The RWPF1 introduces the highest flow of water into the network compared to the other two pumping facilities, approximately 200 L per second. It consists of four wells with a submerged intake structure, featuring four electronically controlled pumps automatically driven by local logic based on the piezometric level in the storage tank, shared with RWPF2, where the water is stored.

The RWPF1 includes a coal filtration system responsible for removing nitrates. The filtration system consists of a filter that removes nitrates from the water and a brine tank. The brine tank plays its role in the treatment of water hardness, where special resins, usually sodium cycle resins, are used. These resins have the issue of reaching saturation after a certain treatment period, and to be reused, they need to be regenerated. The brine, which is nothing more than a concentrated salt solution, is precisely used for this regeneration process. From a chemical standpoint, it simply removes the calcium fixed in the resins and recharges them with sodium.

A computing system based on PLC is present to control the operations of the treatment system. Figure 17 shows the main components of the RWPF1.

In order to represent RWPF1, the AAS_RawWaterPumpingFacility type described in Section 5.2.3 has been considered, adapting the Components Submodel in order to include a reference to the AAS of the water treatment system.

Figure 18, on the left, shows the structure of the AAS modeling RWPF1. As can be seen, the Components Submodel includes the SMC “WaterTreatmentFacilities”, which contains the ReferenceElement to the AAS modeling the nitrate filtration system, as shown in Figure 18. To model the nitrate filtration system, an instance of the AAS_WaterTreatmentFacility type, described in Section 5.2.5, was created. This instance has been called “NitrateTreatmentFilter_RWPF1”, and it is shown on the right side of Figure 18.

Considering the instance NitrateTreatmentFilter_RWPF1, in the Documentation Submodel, important documentation has been included. The submodel named “NitrateFilter” plays a fundamental role within the overall system, providing a detailed representation of the information related to the nitrate treatment phase within the plant. Let us analyze some of the fields and properties visible in Figure 18. The “FilterType” field specifies the type of filter used for nitrate treatment. In our case study, there is a reference to an “Ion Exchange Resin Filter”, representing a common technology for addressing nitrate presence in the water. The “FiltrationEfficiency” property quantifies the percentage of the filter’s efficiency in removing nitrates from water; in the case study, the efficiency is 95%, indicating a high degree of purification. The “Capacity” property indicates the maximum amount of water the filter can treat in one minute. The “Status” property reflects the current condition of the filter; in Figure 18, the filter is indicated as “Operational”, but other states such as “Under Maintenance” or “Out of Service” may exist depending on operational circumstances. The “ResidualCapacity” property determines the residual capacity of the filter; the replacement threshold is defined as 20% of the residual capacity, indicating the need to replace the filter to ensure optimal operation. Considering the Components Submodel, among the components, Figure 18 shows the pointer to the AAS modeling the brine tank; an AAS_Tank has been used to model it, describing the characteristics of the brine tank and its basic properties.

## 8. Final Remarks

This paper has provided a detailed overview of the fundamental role of the Asset Administration Shell in the context of creating digital twins for real systems. Starting from the analysis of the current literature, it has been pointed out that there is a lack of proposals about the use of an Asset Administration Shell to model water supply systems. This paper has proposed the definition of several Asset Administration Shell types to represent the complex components that make up a water resource management system from the water source to the final distribution.

This work was preceded by an exhaustive study of all the features of a WSS, in order to define a model including all of them that was also capable of representing any real WSS. The verification of the capability of the AAS information model to represent any WSS was considered a very important issue. For this reason, it was applied to several case studies. As this work was conducted inside a research project funded by the Italian Ministry of Enterprises and Made in Italy, partners of this project produced several real case studies on which the model has been applied. Among the several case studies, there is that presented in this paper. The application of the AAS model to the considered real scenarios allowed us to verify the feasibility of the customization of the model to a real WSS featuring particular and unique constraints and features. The AAS information model has been realized and uploaded on the GitHub platform in order to allow its use and testing in real scenarios by other researchers.

## Figures and Tables

**Figure 1 sensors-24-01360-f001:**
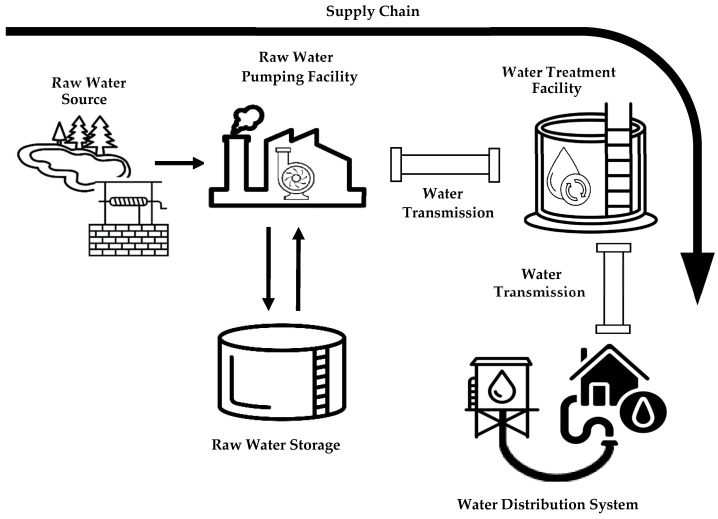
Subsystems of a water supply system along the water supply chain.

**Figure 2 sensors-24-01360-f002:**
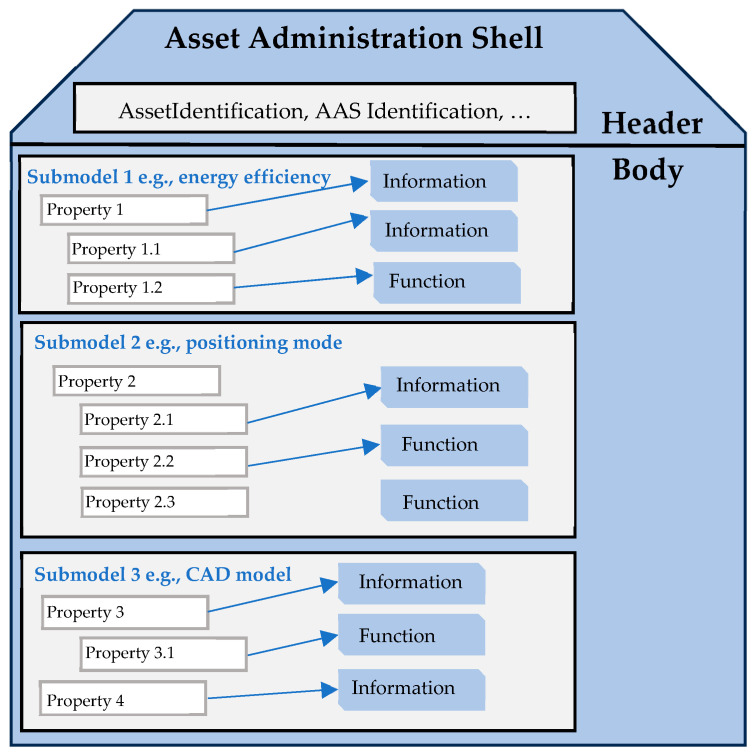
Structure of an AAS metamodel.

**Figure 3 sensors-24-01360-f003:**
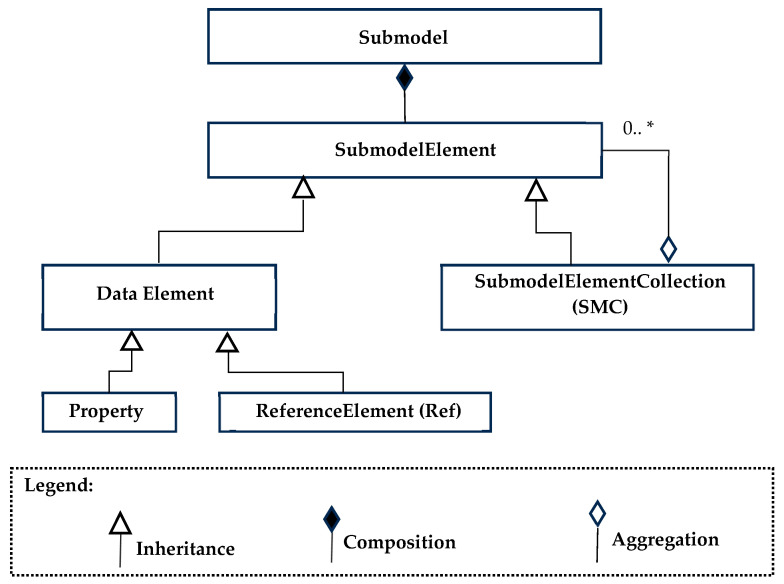
Class hierarchy relevant to the submodel structure.

**Figure 4 sensors-24-01360-f004:**
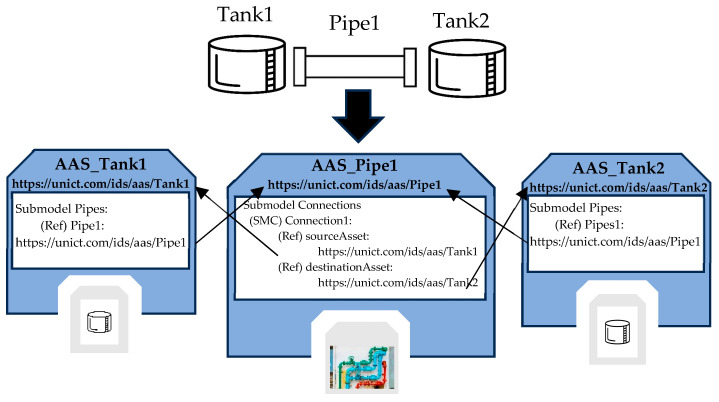
Example of the use of the AAS_Pipe and AAS_Tank models.

**Figure 5 sensors-24-01360-f005:**
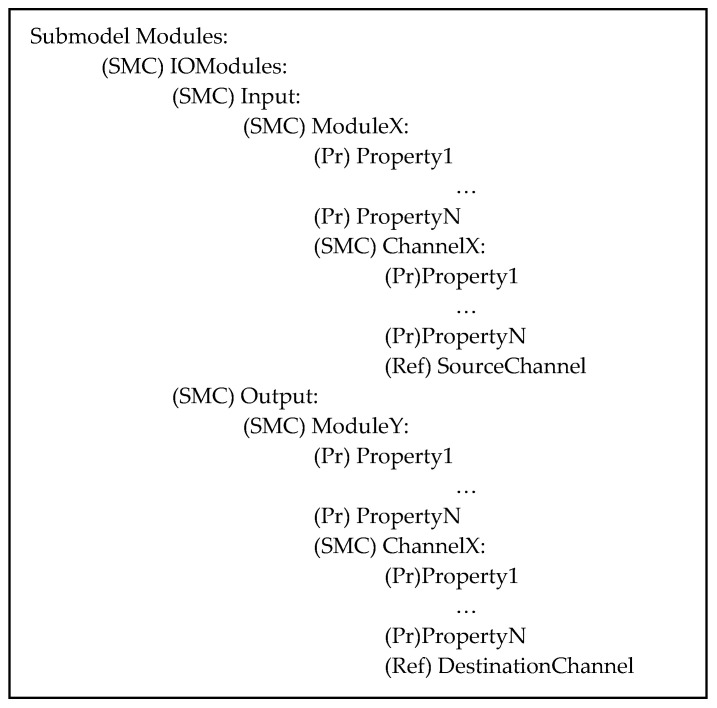
Structure of the Modules Submodel of the AAS_SMC model.

**Figure 6 sensors-24-01360-f006:**
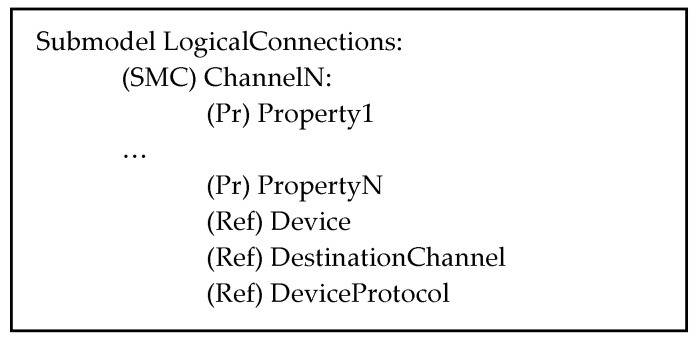
Structure of the LogicalConnections Submodel of the AAS_CS model.

**Figure 7 sensors-24-01360-f007:**
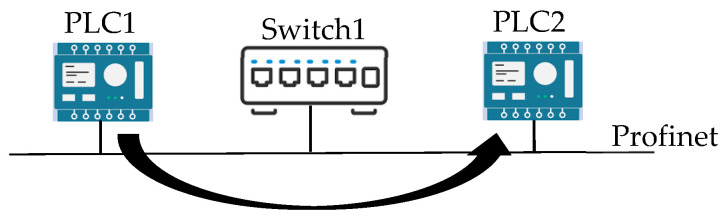
Example of a communication system made up of two PLCs connected through a switch using the Profinet communication protocol.

**Figure 8 sensors-24-01360-f008:**
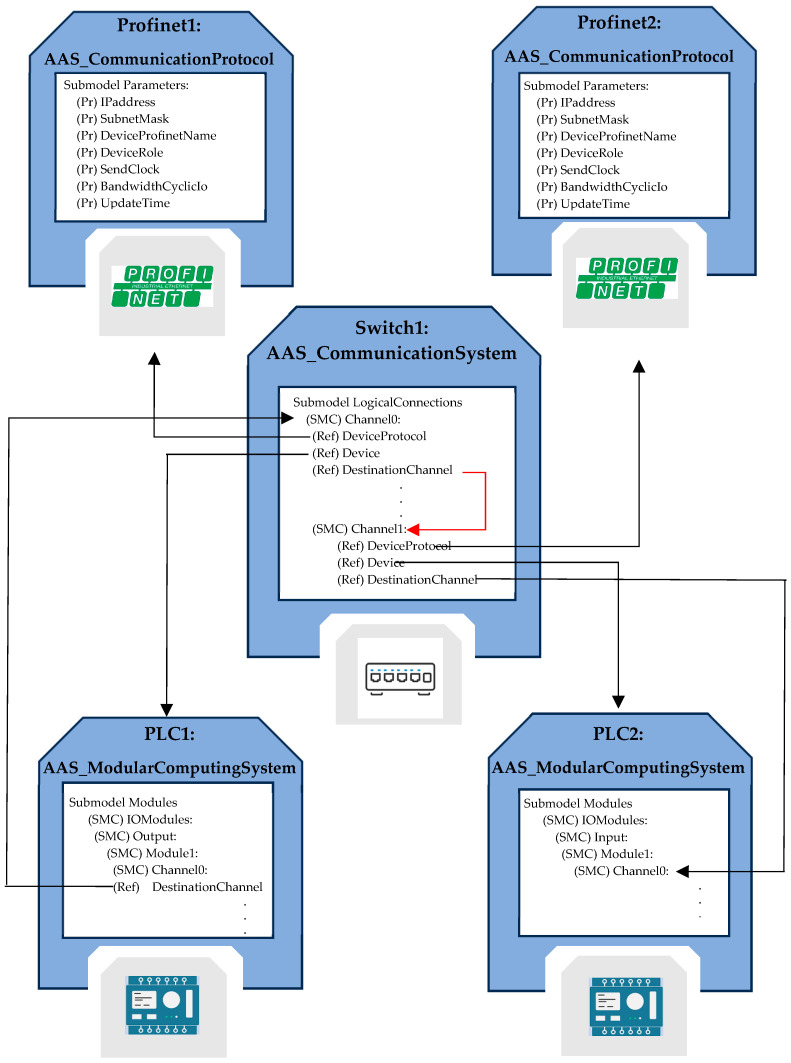
Representation by the AAS of the communication scenario that is depicted in Figure 7.

**Figure 9 sensors-24-01360-f009:**
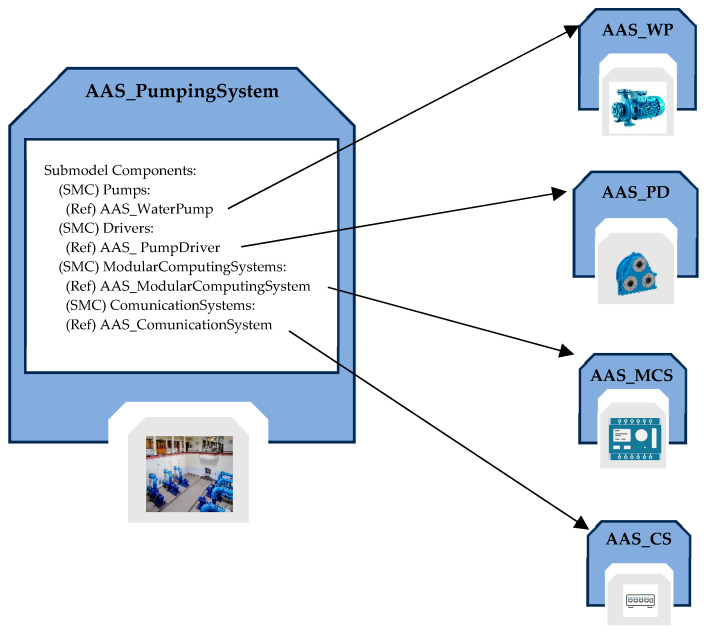
Structure of the Components Submodel inside the AAS modeling the pumping system.

**Figure 10 sensors-24-01360-f010:**
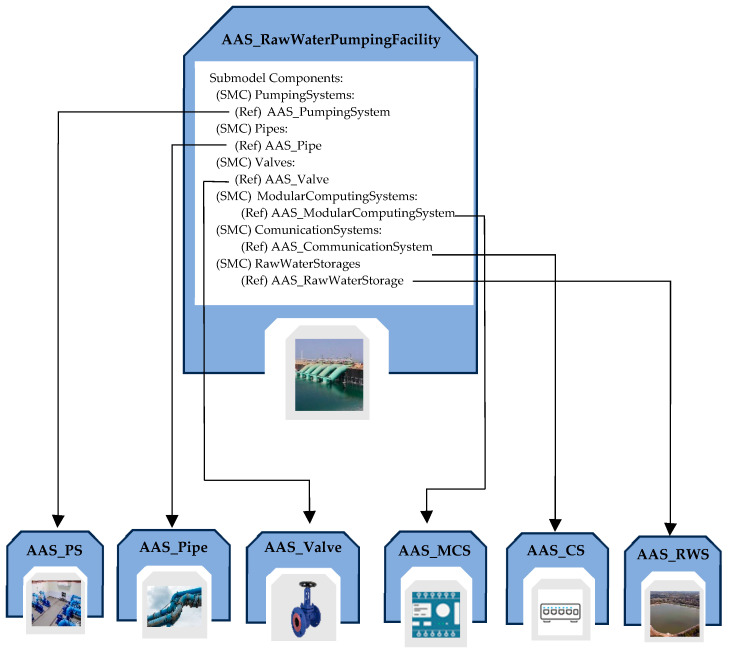
Structure of the Components Submodel inside the AAS modeling the raw water pumping facility.

**Figure 11 sensors-24-01360-f011:**
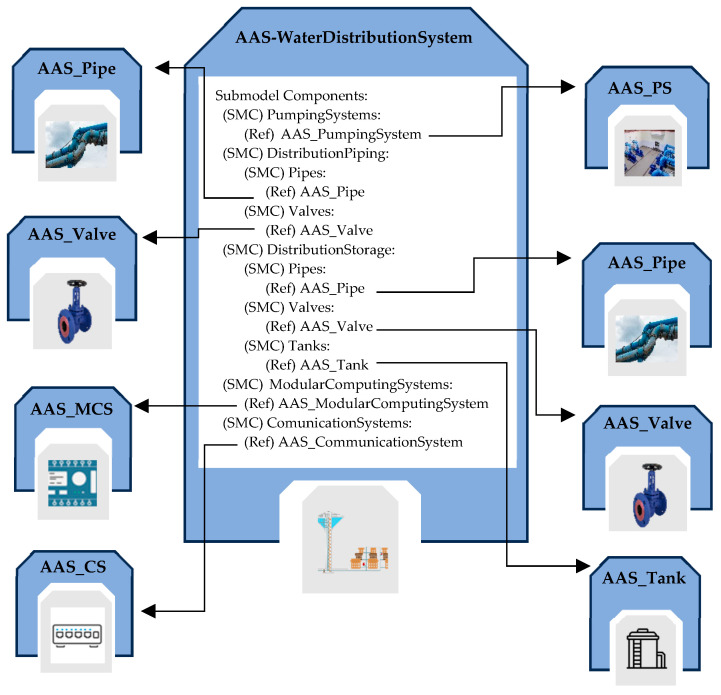
Structure of the Components Submodel inside AAS modeling the water distribution system.

**Figure 12 sensors-24-01360-f012:**
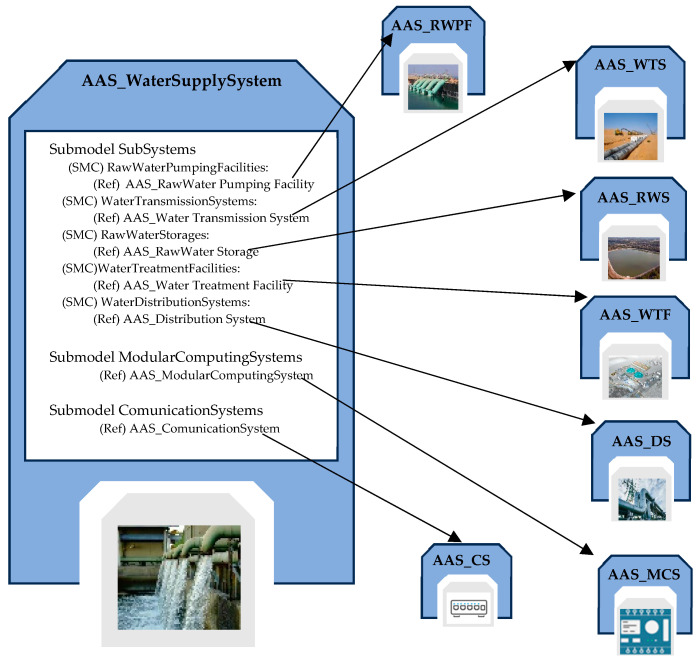
Structure of the SubSystems Submodel inside the AAS modeling the water supply system.

**Figure 13 sensors-24-01360-f013:**
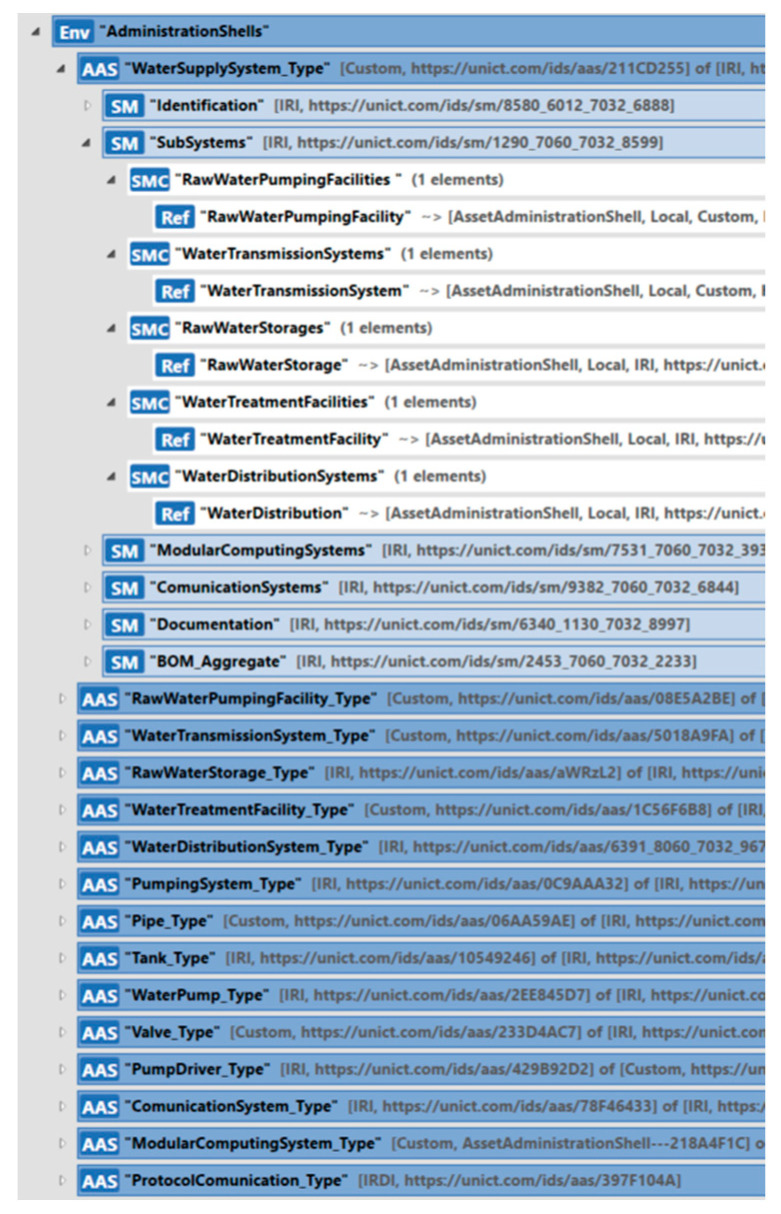
Implementation of the AAS_WaterSupplySystem by the AASX Package Explore.

**Figure 14 sensors-24-01360-f014:**
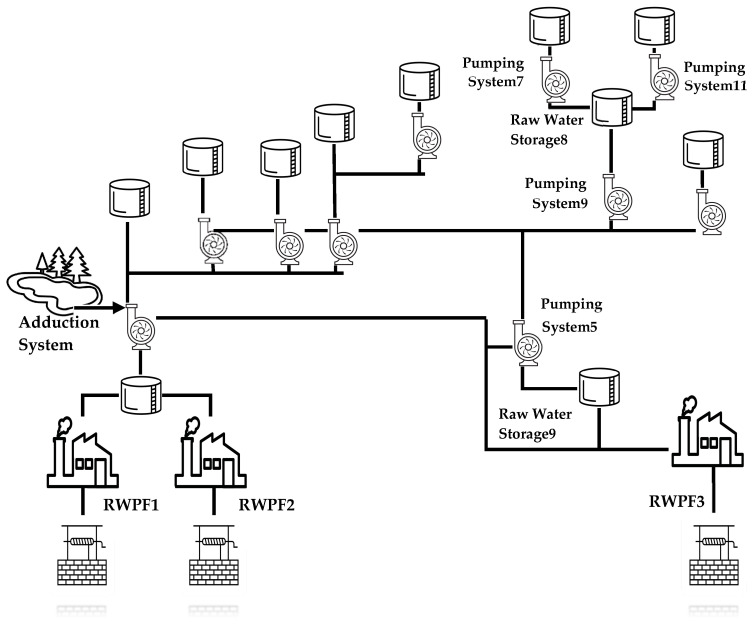
Structure of the water supply system considered for the case study.

**Figure 15 sensors-24-01360-f015:**
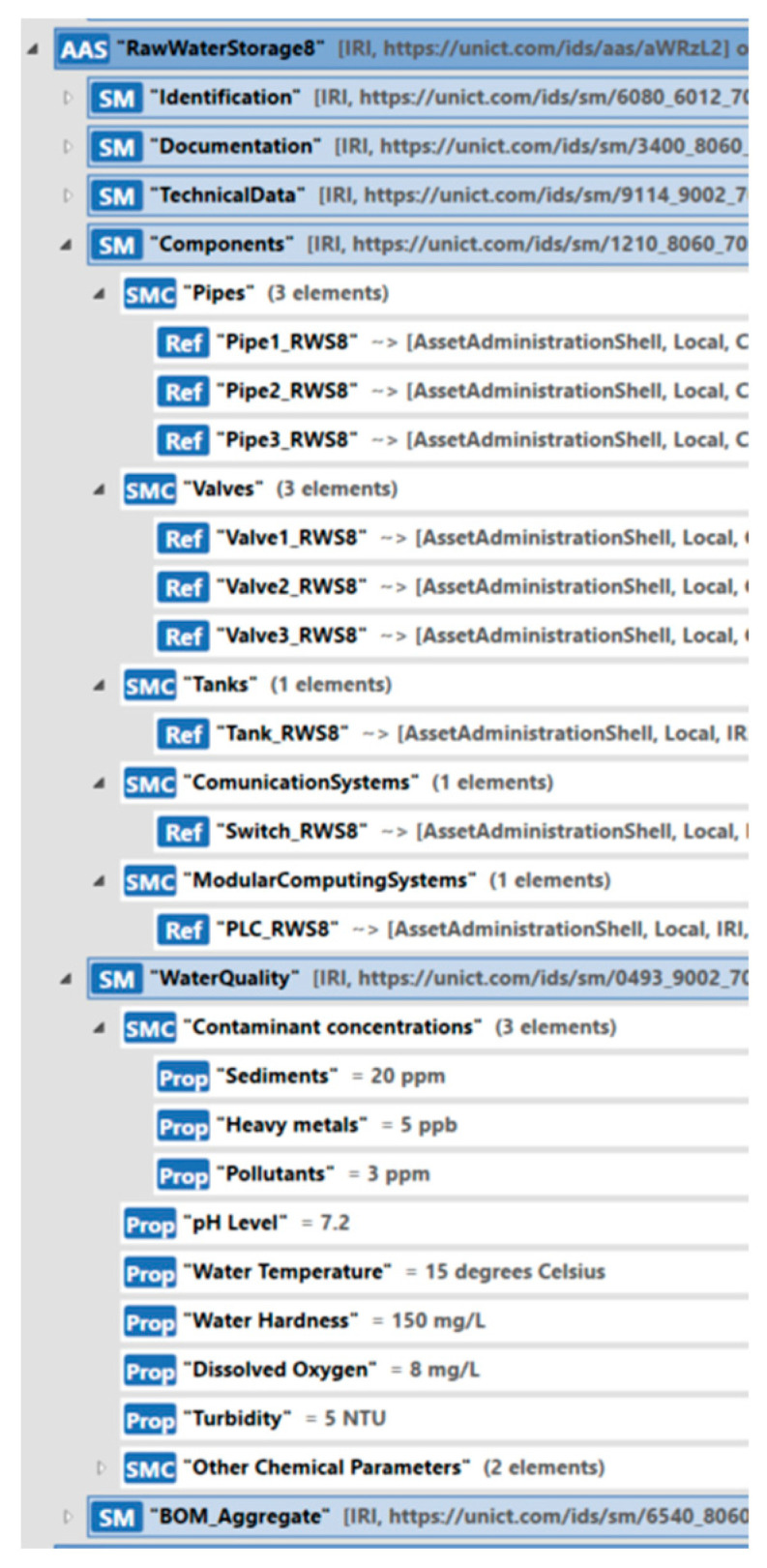
AAS model of RawWaterStorage8.

**Figure 16 sensors-24-01360-f016:**
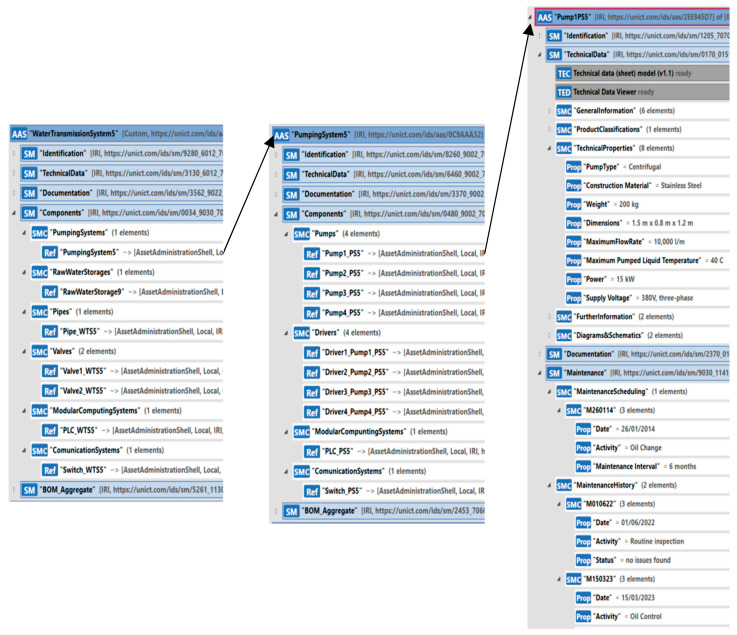
Instance of AAS_WaterTransmissionSystem type.

**Figure 17 sensors-24-01360-f017:**
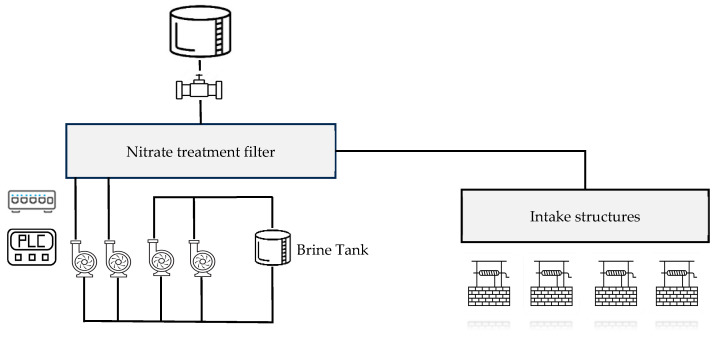
Main components present in the RWPF1.

**Figure 18 sensors-24-01360-f018:**
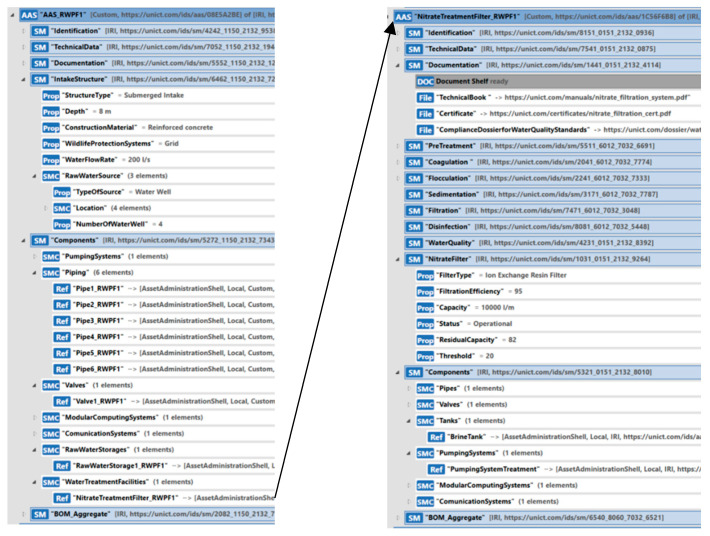
AAS instance of the AAS_RWPF type modeling the RWPF1 including the nitrate treatment filter.

## Data Availability

The data and models presented in this study are openly available in https://github.com/OPCUAUniCT/AASWaterSupplySystem.

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
