# Peer review of "Digital Twin of a Water Supply System Using the Asset Administration Shell"

_sensors, 2024, doi:10.3390/s24051360_

Round 1

Reviewer 1 Report

Comments and Suggestions for Authors

This manuscript outlines a comprehensive information model of a water supply system based on AAS submodel components. The paper is well written and the authors describe the various components very well.

However, the introduction makes some claims that are not backed by the actual content of the paper. For instance, the introduction mentions simulation, detection of anomalies, etc. It is not clarified, though, that the presented AAS submodel structure is just the first step supporting this. Therefore, I'd like the authors to add such a clarification. 

Moreover, it appears as if the authors defined all submodel components on their own. I would have expected that for some elements submodel components were already defined and published. 

In addition, the authors could mention how the submodel structure was validated.

Reviewer 2 Report

Comments and Suggestions for Authors

1. In the Line 64, the authors claimed that the term AAS is used instead of the term DT within the Industry 4.0 context. Basically, digital twin is a loose metaphorical concept, while AAS is a description framework. Therefore, the authors should argue more about the logic relation between Digital Twin and AAS. Maybe the following paper can help to argue this issue.

"Petri nets-based digital twin drives dual-arm cooperative manipulation." Computers in Industry 147 (2023): 103880.

2. The paper lacks the introduction of the contents of Identifiers. Because different assets need a unique identification, and identifiers are the basic units to formally describe AAS.

3. The essence of AAS is digital information model, so the semantic dictionary and standardization of information model are indispensable parts.

4. This paper uses IRI, but does not use IEC,ISO,IRDI and other external dictionary standards to construct AAS, so there is a lack of standardization of concepts.

5. In the hierarchical description of AAS, the author uses the method shown in Figure 3. Here, it is recommended to add other description methods such as UML class diagram to further express the structure in AAS more intuitively and concretely.

6. In section 4.4, the author uses IDTA's AAS submodel template to model the water supply system semantically. However, in the actual process, according to the specific assets, the template is not unique. For example, the document submodel mentioned in this paper has multiple versions in the development tool.

7. In Figure 4, which uses SMC to build submodels, the connection relationship between submodels is not fully reflected. For example, the id of connecting AAS can be added after ref to better reflect a nested relationship.

8. This paper is a digital twin of water supply system based on AAS, but the content of detection, management and simulation is not reflected in this paper.

9. This paper is a digital twin of water supply system based on AAS, but the content of detection, management and simulation is not reflected in this paper. Does not reflect the advantages of AAS for digital twins.

10. In the last example, the collaborative aspect of water supply system data is not well reflected.

To sum up, it seems that this research is not in-depth academically. The research gap of current , academic contributions and research limitations should be highlighted. 

Comments on the Quality of English Language

Good

Round 2

Reviewer 2 Report

Comments and Suggestions for Authors

1. According to lines 54 to 64, introducing DT has many advantages. Can the model presented by the author achieve the above functions? Compared with the advantages of these ideas, what are the gaps? It is best for the author to further clarify the limitations of this study.

2. To highlight the importance and contributions of this research, more ten references papers about AAS(you will find more)are suggested to be cited and introduced.

Comments on the Quality of English Language

General
